# Enhancing Functional Ability in Chronic Nonspecific Lower Back Pain: The Impact of EMG-Guided Trunk Stabilization Exercises

**DOI:** 10.3390/healthcare11152153

**Published:** 2023-07-28

**Authors:** Shivani Porwal, Moattar Raza Rizvi, Ankita Sharma, Fuzail Ahmad, Mastour Saeed Alshahrani, Abdullah Raizah, Abdul Rahim Shaik, Mohamed K. Seyam, Mohammad Miraj, Batool Abdulelah Alkhamis, Debjani Mukherjee, Irshad Ahmad

**Affiliations:** 1Department of Physiotherapy, School of Allied Health Sciences, Manav Rachna International Institute and Studies (MRIIRS), Faridabad 121001, India; shivaniporwal230@gmail.com (S.P.); ankitasharma.fas@mriu.edu.in (A.S.); 2School of Allied Health Sciences, Manav Rachna International Institute and Studies (MRIIRS), Faridabad 121001, India; mrrizvi.fas@mriu.edu.in; 3Respiratory Care Department, College of Applied Sciences, Almaarefa University, Diriya, Riyadh 13713, Saudi Arabia; 4Department of Medical Rehabilitation Sciences, College of Applied Medical Sciences, King Khalid University, Abha 61413, Saudi Arabia; msdalshahrani@kku.edu.sa (M.S.A.); balkamees@kku.edu.sa (B.A.A.); debjani@kku.edu.sa (D.M.); 5Department of Orthopaedics, College of Medicine, King Khalid University, Abha 61413, Saudi Arabia; araizah@kku.edu.sa; 6Department of Physical Therapy & Health Rehabilitation, College of Applied Medical Sciences, Majmaah University, Al Majma’ah 15431, Saudi Arabia; s.abdulrahim@mu.edu.sa (A.R.S.); m.seyam@mu.edu.sa (M.K.S.); m.molla@mu.edu.sa (M.M.)

**Keywords:** chronic nonspecific lower back pain, trunk rotation, trunk-stabilization exercise, electromyography biofeedback, muscle strength, lifestyle measures, behavior modification

## Abstract

Background: Nonspecific lower back pain (NSLBP) is described as pain that is not caused by an identifiable, well-known disease, such as infection, tumor, osteoporosis, fracture, structural deformity, inflammatory condition, radicular syndrome, or cauda equina syndrome. Aim: The aim of this study was to determine the effect of EMG-guided trunk stabilization exercises on functional disability associated with LBP. Materials and Methods: A single-blinded pre- and post-test experimental comparative design was used for this study. Fifty individuals with chronic NSLBP were screened for inclusion criteria. Of these, forty were randomly grouped into the EMG group receiving trunk-stability exercises with electromyography biofeedback and non-EMG group receiving trunk-stabilization exercises without EMG biofeedback. Participants performed five trunk-stability exercises 3 days a week for 4 weeks. The intensity of pain, range of motion, functional disability, and balance were measured at baseline and after 4 weeks. Results: Both techniques indicated a significant effect on chronic NSLBP; however, trunk-stability exercises combined with EMG biofeedback produced better results in alleviating the intensity of pain, increasing the range of motion, and improving functional disabilities and static balance. Conclusion: The present study confirms that trunk-stability exercises with EMG biofeedback can be practiced safely, contributes to a greater boost in neuromuscular efficiency in the lumbar flexors and extensors, and is effective in modifying functional disability for patients with NSLBP.

## 1. Introduction

Urban society has a high rate of chronic lower back pain (LBP), which hinders people from working and limits their daily activities [1]. This condition has a huge impact on health, social services, and economics in the community. Compared to Saudi Arabia, Western societies are experiencing some of the most severe forms of suffering, disability, and enormous economic costs, which include frequent medical consultations, rehabilitation, and hospitalizations [2,3].

According to a survey, 42.4% of people suffer from LBP each year, and 22.8% suffer from it each week. Approximately one-fifth of patients (20.6%) experienced restrictions in everyday activities because of LBP, and 14.4% reported feeling emotionally unhappy because of LBP [4]. Nonspecific lower back pain refers to pain in the lower back region that is not caused by a specific underlying condition such as osteoporosis, fracture, structural deformity, infection, tumor, inflammatory condition, radicular, or cauda equina syndrome. Nonspecific lower back pain is a common condition and is often attributed to factors such as muscle strain, poor posture, sedentary lifestyle, or mechanical stress on the spine. Chronic lower back pain (CLBP) is related to histomorphology and anatomical alterations in the paraspinal muscles [5]. These back muscles are smaller, have more fat, and have certain atrophic alterations in specific muscle fibers [6]. The reduced activity and strength of these muscles increase the functional disability associated with CLBP.

Symptoms of NSLBP can include cyclic pain, buttock and thigh discomfort, morning stiffness or soreness, discomfort at the commencement of a movement, discomfort in forward flexion and returning to an upright position, increased pain with extension, side flexion, rotation, standing, walking, sitting, and activity in general, ease of pain while changing different postures, and lying down, particularly in the fetal position, to alleviate pain [7].

Numerous activities, such as walking and sports, need trunk rotation. One of the main risk factors for lower back discomfort is trunk rotation. The latissimus dorsi, internal oblique, and transversus abdominis work ipsilaterally during rotation, whereas the external oblique, rectus abdominis, and lumbar multifidus work contralaterally [8]. Mechanical lower back pain includes conditions including lumbosacral muscle and ligament injury, arthritic facet joints or sacroiliac joints, and degenerative disc disease [9]. Joint support and stability are the primary functions of the local muscles. They do not usually create movement, but they do give the stability that allows a joint to move. 

Chronic lower back pain alters the lower back’s structure and surrounding tissues, leading to weakened abdominal muscles responsible for trunk stability. This causes discomfort and functional limitations. As the pain worsens, physical activity is restricted, and the muscles around the spine may atrophy, reducing in size. Spinal instability is the most prevalent biomechanical cause of lower back pain. Chronic lower back pain sufferers perceive lumbar vertebral instability as a severe problem. It causes discomfort, decreases endurance and flexibility, and restricts the lumbar range of motion [10]. The trunk muscles are most important to maintain spinal stability. The spinal column cannot carry normal loads if the trunk muscles are not providing enough support. 

Balance is dependent on the interplay of the vestibular, visual, and proprioception systems [11]. The correct functioning of these systems may be jeopardized in people with CLBP. Certain studies have revealed a relationship between low lumbosacral proprioception and impaired balance in those with CLBP. Furthermore, changes in motor cortex anatomy have been associated with impaired postural control in patients with persistent lower back pain [12]. Sensory tissue damage in the lumbar spine, trunk, or lower extremities may jeopardize postural stability on a fundamental level. The loss of proprioceptive information in these places might be the deciding factor in sensory integration becoming less reliable [13].

Muscle recruitment varies based on the activity performed. Lifting a lighter weight involves fewer muscles, while lifting a heavier weight requires more muscles to generate force. As force increases, the central nervous system becomes more stimulated, activating additional motor units and increasing their firing rate. This leads to larger EMG signal amplitudes, indicating heightened muscle activity. By using electromyography, we may exert force, make movements, and perform many other actions that let us engage with the environment. The electromyograph is a bioelectric signal that has a wide range of uses that have grown over time. Neurological problems are typically diagnosed using electromyography in clinical settings. It is widely used to evaluate individuals with neuromuscular illnesses, lower back pain, and motor control abnormalities [2]. In addition to physiological and biomechanical research, EMG has been developed as an assessment tool in applied research, physiotherapy, rehabilitation, sports medicine and training, biofeedback, and ergonomics research [14]. Biofeedback and EMG are useful tools for diagnosing and treating neuromuscular disorders, lower back pain, and motor control problems. EMG biofeedback-assisted trunk stabilization exercises may improve trunk muscle activity, strength, proprioception, and balance control in patients with CLBP. Therefore, this study aims to reduce the societal cost of CLBP while improving patients’ functional outcomes and quality of life by examining the effects of EMG biofeedback-guided trunk stabilization exercises on CLBP symptoms such as pain, balance, lumbar range of motion, and functional impairment.

## 2. Materials and Methods

### 2.1. Study Design

This was an experimental pre–post-test comparative study comparing the effects of trunk-stability exercises with and without electromyography biofeedback in patients with nonspecific chronic lower back pain.

### 2.2. Sample Size Calculation

The sample size was calculated using a priori power analysis based on our primary outcome, specifically the pain level at a 6-week follow-up. This analysis utilized data from a previous study that investigated the efficacy of electromyography biofeedback training on trunk stability [2], resulting in an effect size (d) of 0.93. We then used G*Power version 3.1.9.7 software to calculate the sample size based on an independent student *t*-test with a statistical significance level of 0.05. The results of the power analysis indicated that a sample size of 40 participants was necessary to achieve statistical significance with a power of 80%: critical t = 2.02, D = 2.94, and power = 0.818.

### 2.3. Participants

Fifty participants, aged from 20 to 45 years, at the university campus, were screened for the inclusion criteria in the study. In this study, convenience sampling was employed to select participants based on their availability and accessibility, allowing for a quick and convenient data-collection process. Participants were eliminated if they had been diagnosed with a tumor, infection, inflammatory disease, spinal or lower limb surgery, spinal fractures, a manifestation of radiculopathy (showing signs of nerve root compression), severe osteoporosis, structural deformities such as spondylolisthesis, or an incompatibility with exercise therapy (uncontrolled HTN, previous myocardial infarction, cerebrovascular disease, peripheral vascular disease, respiratory disorder). Ten participants were excluded as they failed to comply with the inclusion criteria or declined to continue to participate. The remaining 40 participants were randomly allocated to either the non-EMG or EMG group, each containing 20 participants (Figure 1). A balanced randomization procedure was employed to assign participants to different treatment groups. This involved the selection of sealed envelopes from opaque envelopes containing group assignment numbers, which were generated using a computer random number generator. The study employed a single-blinded design where the participants were unaware of their group assignments. However, the researchers who conducted the study, including those responsible for electrode placement, administering the exercise, and assessing muscle contractions for biofeedback, were aware of the group assignments. Both groups received a trunk-stabilization exercise with electrode placement, but only one group received EMG biofeedback during the exercise. By implementing this design, the study aimed to minimize bias by ensuring that participants remained unaware of their group assignments while allowing the researchers to accurately assess the effects of EMG biofeedback. By comparing the outcomes between the two groups, the study aims to determine the specific effects of EMG biofeedback on the trunk-stabilization exercise, allowing researchers to evaluate its potential benefits compared to performing the exercise without feedback. The study was approved by the ethical committee at the Faculty of Allied Health Sciences and complied with ethical principles for medical research involving humans (WMA Declaration of Helsinki).

### 2.4. Experimental Procedures and Measurements

All participants signed a consent form before taking part in this study. Baseline measurements were obtained for the intensity of pain using VAS, static balance through the APP-Coo test, and range of motion through modified Schober’s test [15]. In addition to this, the Oswestry Disability Index questionnaire was filled out by each participant. The data were collected at Day 0 and after 4 weeks of duration. 

The APP-Coo test is an application that can be used on smartphones and tablets and is based on a triaxial accelerometer. It can assess both static and dynamic balance deficits [15]. The APP-Coo was used to assess the capacity to balance by placing a Samsung Galaxy A51 (Android 12) on the breastbone and immobilizing it with an elastic band. The application perceives trunk oscillations using a triaxial low-g acceleration sensor, which provides acceleration measurements on three perpendicular axes. This test may evaluate the trunk’s oscillation in several static situations, including “feet together” and “on a broad base” (30 cm distance between both malleoli). A 5-s countdown begins once the patient presses the application’s start button, preparing the patient to perform the test. After the test, a red dot that moves across a virtual grid provides a quantitative evaluation of the body sways that occurred in the longitudinal, frontal, and transverse axes while the participant was performing the task. The patient was given instructions to stand up and maintain stability while keeping their eyes open or closed, feet together, or on a broad base. For the patient’s protection, the evaluation room had adequate lighting, and the floor was treated to prevent slipping. 

The lumbar spine’s range of motion was measured using modified Schobers’s test and the tape method was used for all other motions, i.e., flexion, extension, and lateral bending [8]. The measurement was taken thrice, out of which the best of the three was taken as the final reading. 

In hospital settings, the Oswestry Lower Back Disability Index (ODI) is the most applied outcome-measure questionnaire for assessing pain in the lower back [16]. This is a structured and self-administered questionnaire with 10 parts meant to identify limits in various everyday tasks. Each segment is graded on a scale from zero to five, with five indicating the most disability. When calculating the index, the total score is divided by the highest possible score, that number is multiplied by 100, and the resulting value is expressed as a percentage. Therefore, each unanswered question reduces the denominator by five. If a patient selects several responses to a question, the highest-scoring response is recorded as a real sign of impairment. Further, the disability level is categorized based on percentage into minimal disability (0–20%), moderate disability (21–40%), severe disability (41–60%), crippled (61–80%), and completely disabled (81–100%). The questionnaire requires around 5 min to finish and approximately 1–2 min to score [17,18].

Figure 1 shows the flow chart of the study design. Following the completion of baseline evaluations, participants were entered into a lottery in which they were randomly assigned to either the EMG group or the non-EMG group. Both groups were given trunk-stability exercises; however, the EMG group received EMG biofeedback in addition, whereas the non-EMG group merely participated in trunk-stabilization exercises. The exercise time for each step in a session was 5 min and the rest time was 2 min. The exercise was performed 3 times a week for 4 weeks, accounting for 12 sessions with each session of about 30–40 min. For every participant, data from three trials were gathered, and an average measurement of the three was recorded.

The investigator supervised the sessions, and participants were instructed to report any adverse event, whether or not it was connected to the exercises. During the trial, participants were not allowed to partake in any other physical activity. Before the trunk-stabilization exercise, all of the participants performed warmup exercises, followed by aerobic work (static bicycling for 5 min at a moderate pace) and stretching (hip flexor, hamstring, calf, adductor stretch, and back stretching).

### 2.5. Trunk-Stabilization Exercise Program

Participants in the EMG group performed trunk-stabilization exercises after the electrodes were placed [19,20]. All of the participants were required to conduct exercises with the gym ball and the balance disc in several positions. They were instructed to take their bellies inwards to stimulate the deep muscles before undertaking any of the exercises in any of the positions. The activities in the first stage included (i) holding a gym ball between the knees with the hip and knee at 90 degrees in a supine position, (ii) bridging with the feet on a balance disc, (iii) prone plank with the feet on a balance disc, (iv) four-point kneeling position with alternate hand and leg raise, and (v) sitting on a gym ball with alternate hand and leg lifting [2,3]. Participants followed two sets of each exercise with an 8-s hold for the first and second weeks. Exercises were made more difficult in the second phase of the program by focusing on trunk and limb coordination, optimal trunk stability, and postural and movement pattern improvement. At this stage, the three sets of each exercise with a 10-s hold were followed in the second and third weeks. Participants were given a break of 1–2 min between each type of exercise. Similar trunk-stabilization exercises were carried out in the non-EMG group without the use of EMG biofeedback.

### 2.6. Flexor and Extensor Trunk EMG Activation during Trunk-Stability Exercises 

Before the deployment of the EMG electrodes, standardized skin preparation was performed. This preparation consisted of the removal of hair, a mild abrasion with sandpaper, and the washing of the skin with an alcohol swab. The electrode–skin impedance was less than 10k Ω after this procedure. The surface EMGs of two right back muscles and four abdominal muscles, including the latissimus dorsi, rectus abdominis, external oblique and internal oblique and transversus abdominis, were measured at a sampling frequency of 1024 Hz employing hydrogel and disposable Ag/AgCl electrodes measuring 38 mm by 19 mm. In order to gather EMG data and use them as a reference for normalization, three 5-s trials of manually resisted maximal voluntary contraction (MVC) were conducted on each muscle. EMG signals from both the upper and lower rectus abdominals were recorded, along with signals from the upper rectus abdominis (located 3 cm lateral and 5 cm superior to the umbilicus), lower rectus abdominis (located 3 cm lateral and 5 cm inferior to the umbilicus), upper back extensors (located 2 cm lateral to the midline running through the T9 spinal process), and lower back extensors (located 2 cm lateral to the midline running through the L5 spinal process) [8,21].

### 2.7. Data Analysis

Statistical analysis was performed with the help of SPSS Version 22. Before beginning parametric tests, the assumption of normality was evaluated using a Shapiro–Wilk test. The data distribution for all variables at all levels was tested at *p* < 0.05. An independent *t*-test was used to analyze the data between two groups followed by a paired *t*-test to analyze the data within each group. The confidence interval was set to 95%, while the level of significance was set to *p* < 0.05. Descriptive statistics were used for analysis and determination of the mean and standard deviation of participants with and without EMG training.

## 3. Results

In total, 10 individuals did not meet the inclusion criteria, resulting in a final sample size of 40 participants (*n* = 40). These participants were then equally divided into two groups based on the intervention received, each consisting of 20 participants. The non-EMG group received trunk-stabilization exercises without EMG biofeedback, while the EMG group received trunk-stabilization exercises with EMG biofeedback. A pre-assessment of different outcome variables was performed in the non-EMG group (*n* = 20) and the EMG group (*n* = 20). However, during the post-test measurement at the four-week follow-up, two participants from the non-EMG group and three participants from the EMG group did not attend; these participants were removed from the study, leaving 17 participants in the EMG group and 18 in the non-EMG group (Figure 1).

Table 1 presents the demographic and clinical characteristics of participants in the EMG and non-EMG groups. For the outcome measures related to demographics (age, height, weight, and BMI), the mean values with their standard deviations (SD) are provided for both the non-EMG and EMG groups. Regarding the categorical variables, such as marital status, educational qualifications, duration of NSLBP, and use of medications (nonsteroidal anti-inflammatory drugs; NSAIDs), the table presents the frequencies (*n*) and percentages (%) for each category within the EMG and non-EMG groups. Additionally, for these categorical variables, a chi-square test was performed to assess the level of independence between the groups. 

An independent sample *t*-test was performed to compare the nominal demographic variables between the non-EMG and EMG groups, which showed no significant difference. Furthermore, the independent sample *t*-test test was applied on different outcome variables to compare the pre- and post-effect of trunk-stability exercises with and without EMG biofeedback in chronic lower back pain patients (Table 2). There were significant differences in pain intensity, functional disability and flexion between the non-EMG and EMG groups during the post-test measurement. In addition, the pre-assessment of different outcome variables like pain, functional disability, range of motion, and static balance showed no significant differences between the non-EMG and EMG groups.

A paired *t*-test was performed to compare the pre- and post-test effects of trunk-stability exercises with and without EMG biofeedback in chronic lower back pain patients (Table 3). The paired *t*-test showed a significant improvement (*p* < 0.001) in pain as reflected by the VAS score and in function disability as obtained from the ODI% in both groups. However, the mean difference in functional disability was on the higher side (12.58 ± 8.59) in the EMG group as compared to the mean difference (8.33 ± 10.13). Regarding the range of motion, both the flexion and extension of the EMG group improved significantly (*p* < 0.001). In addition to this, the EMG group showed a significant improvement in static balance as measured through the APP-Coo test with closed eyes (*p* = 0.01) which was not observed in the non-EMG group.

Figure 2 presents the EMG activity of trunk flexors and extensors during trunk-stabilization exercises at baseline, 2 weeks, and 4 weeks. The exercises showed varying patterns of EMG activity over time. In the “ball sitting with alternate hand and leg raise” exercise, both flexors and extensors exhibited increased activity from baseline to 2 weeks and further increases at 4 weeks. The “bridging on a balance disc” exercise showed stable flexor activity across the three time points. The “gym ball hold between the knees in supine” exercise demonstrated a gradual increase in flexor activity over the study period. The “prone plank on a balance disc” exercise showed a notable increase in flexor activity from baseline to 2 weeks and a further increase at 4 weeks. The “four-point kneeling with alternate hand and leg raise” exercise displayed consistent increases in flexor activity over time. These findings suggest changes in muscle activation during trunk-stabilization exercises, indicating potential improvements or adaptations.

Functional disability was evaluated through the ODI score as shown in Figure 3. In response to different activities measuring functional disability through the ODI questionnaire, trunk-stabilization exercises without EMG showed significant improvements in pain intensity, personal care, social life, and traveling to manage everyday life (Figure 3A). On the other hand, trunk stabilization with EMG showed significant improvements in pain intensity, personal care, lifting, sitting, standing, social life, and traveling (Figure 3B). The baseline measurement of ODI showed that there was no significant difference in the different activities between the non-EMG and EMG groups (Figure 3C).

The disability levels before and after the intervention for both the non-EMG and EMG groups were determined by calculating the total score from the Oswestry Disability Index (ODI), as depicted in Figure 4. Overall, both the non-EMG and EMG interventions resulted in a decrease in disability levels, with a greater proportion of participants achieving minimal disability status after the intervention. The EMG group had a higher number of participants achieving minimal disability status after the intervention, indicating a greater improvement in their disability levels. Additionally, the EMG group had a greater reduction in the number of participants with moderate disability compared to the non-EMG group.

## 4. Discussion

The aim of this study was to assess the effectiveness of trunk-stability exercises with and without electromyographic biofeedback on functional disability in patients affected with nonspecific CLBP. Additionally, the study aimed to investigate the impact of EMG biofeedback as a supplementary tool for trunk-stabilization exercise in enhancing balance, reducing pain, and increasing the lumbar range of motion. The results of this study demonstrated that combining EMG biofeedback with trunk-stability exercises resulted in better improvements in all outcome measures, including pain severity, range of motion, balance, and functional disabilities. Both groups exhibited significant improvements over the four-week intervention period. However, the EMG group exhibited significantly greater improvements compared to the non-EMG group. Furthermore, the functional impairments of patients in the EMG group improved, with all patients reporting minimal disability. The enhanced improvement observed in the EMG group can be attributed to the immediate auditory and visual feedback provided during contraction, facilitated by EMG feedback. This improvement may be associated with the patient’s increased sense of control over their pain, resulting in reduced pain levels [21,22]. Thus, biofeedback promotes active engagement and motivation in patients, which is a crucial aspect of pain management [23].

The findings of this study revealed a significant reduction in pain in both groups following the four-week intervention. However, the group that received EMG training in addition to trunk-stabilization exercises showed more pronounced results when compared to the group that solely underwent trunk-stabilization exercises without EMG training. The enhanced outcome can be attributed to the combined effect of strengthening deep abdominal muscles, improving flexibility, and enhancing balance through EMG training. The substantial improvement observed in both groups can be attributed to increased back endurance, which subsequently reduces spinal instability and leads to pain reduction. This aligns with previous research indicating that improving endurance and promoting lumbar stability both contribute to pain reduction [24]. Additionally, Hartingan et al. suggested that exercise reduces anxiety and fear, thereby aiding in the improvement of back pain [25]. By engaging the thoracolumbar fascia and increasing intra-abdominal pressure, the muscles of the anterolateral abdominal wall assist in stabilizing the lumbar region of the spinal column. When the motion segment is in a neutral position, the lumbar muscles possess the greatest ability to exert dynamic control, which can explain the pain reduction observed after trunk-stabilization exercises guided by the EMG. The use of EMGs helps to accurately contract the required muscles while maintaining the appropriate threshold [26].

The results of this study indicate that both groups experienced improvements in their range of motion but the EMG group exhibited more significant enhancement. Abnormal or defective movement patterns during the spinal range of motion, known as a poor qualitative or quantitative range of motion, signify dynamic lumbar instability and highlight the need for stability training [24]. Based on the findings of this study, trunk-stability exercises primarily targeted the flexor and extensor compartments of the trunk muscles, resulting in enhanced flexion and an extended range of motion.

Individuals with lower back pain generally exhibit reduced static balance abilities compared to healthy individuals, leading to compromised postural stability. Therefore, it is crucial to incorporate exercises that improve trunk stability to decrease shearing stress on the lower back and enable patients to achieve pelvic and trunk stability. The considerable difference observed between the two groups is a positive indication. Trunk-stability exercises are designed to enhance the body’s balance and stability. These exercises promote the simultaneous activation of abdominal muscles and the multifidus which is a fine motor muscle of the spine [10]. By improving core strength through trunk-stabilization exercises, balance is enhanced, thus reducing the risk of biomechanical disturbances and minimizing the impact on the spine, as evidenced by the study findings [27].

From the baseline assessment to the fourth week, there was a notable improvement in EMG amplitudes for all of the exercises, suggesting improved control over the targeted muscles. The statistical analysis revealed that the EMG group demonstrated a superior performance compared to the non-EMG group.

Finally, this study supports the use of trunk-stability exercises combined with EMG biofeedback as an effective approach to improve functional disabilities, alleviate lower back pain, and address other associated symptoms. This intervention led to enhanced core stability and an improved quality of life. The findings align with a pilot study by Shaughnessy M, which demonstrated that lumbar stabilization contributes to improved quality of life and function outcomes in individuals with chronic lower back pain [28]. Additionally, research by Cholewicki and McGill has emphasized the importance of lumbar segmental muscle activity (rigidity) in maintaining spinal stability in vivo [29]. They highlighted the necessity of motor control to coordinate the activation of the major and small trunk muscles during functional motions. These concepts, along with Bellini’s categorization of trunk muscles into local and global categories, support the notion that deep-back exercises are crucial for achieving segmental stabilization and directly controlling the lumbar region [30,31].

There were a few limitations of the study. Firstly, the sample size was relatively small, which may limit the generalizability of the findings. A larger and more diverse sample would enhance the statistical power and external validity of the study. Secondly, the study only included participants of a specific gender, which restricts the generalizability of the findings to the broader population. Including both genders in future research would allow for a more comprehensive understanding of the effects of the intervention across diverse populations and help to avoid potential gender-related biases or differences in treatment responses. Thirdly, the four-week intervention period might be considered short for assessing long-term effects. Extending the duration or including a follow-up period would provide insights into the sustainability of the observed improvements. In addition, core strength was not assessed and this omission hindered the evaluation of the participants’ functional abilities and potential improvements resulting from the intervention. Evaluating core strength would have provided a more comprehensive understanding of the impact of trunk-stability exercises with EMG biofeedback on overall core strength and its relevance to managing chronic lower back pain.

## 5. Conclusions

This study concludes that combining electromyographic biofeedback with a trunk-stabilization exercise program resulted in remedying functional disabilities and, in turn, a better quality of life, better static balance, improvement in the range of motion, and a reduction in pain severity. However, further research is needed to investigate the long-term effects of this intervention.

### 5.1. Future Scope

This study can be taken further by recruiting equal numbers of males and females in both the groups. The EMG-guided activity can improve the endurance, central obesity, and posture which could also be studied. Patients with lumbar radiculopathy could also be recruited in future studies.

### 5.2. Clinical Application

This study concludes that EMG training is superior to non-EMG training. As a result, EMG training helps with pain, range of motion, functional impairment, and static balance. EMG training gives patients audio-visual feedback on their contractions, which stimulates them to perform well. In individuals with nonspecific chronic lower back pain, EMG training can increase core strength and improve quality of life.

## Figures and Tables

**Figure 1 healthcare-11-02153-f001:**
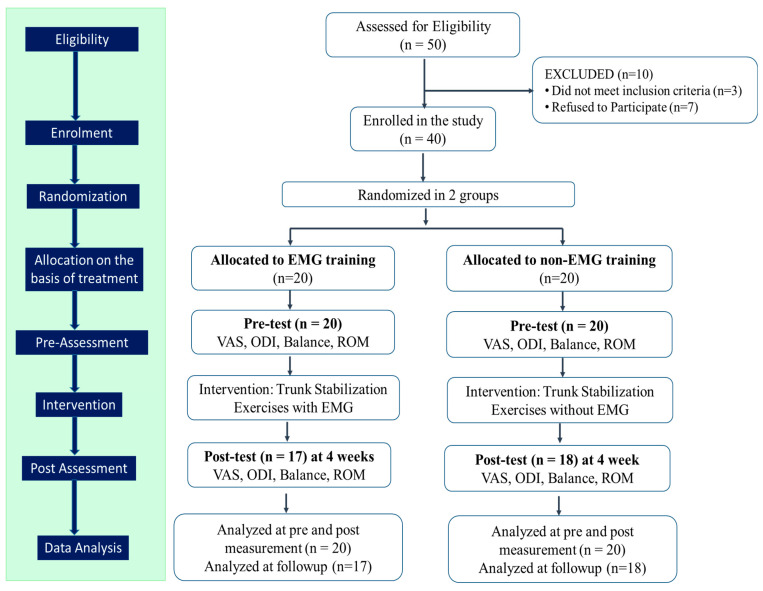
Flow chart of the study design.

**Figure 2 healthcare-11-02153-f002:**
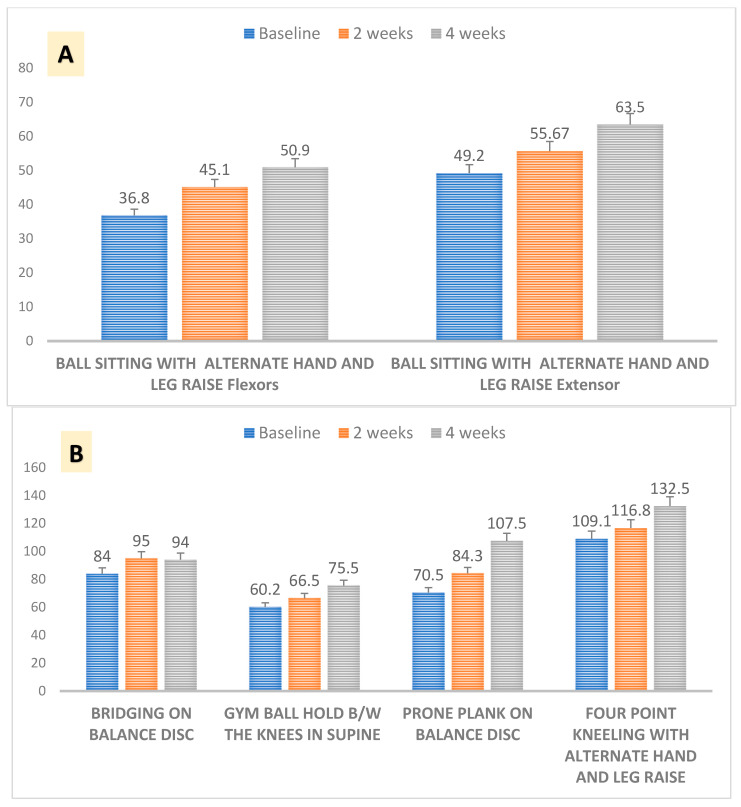
Comparison of EMG amplitudes from baseline to fourth week in flexors and extensors during different trunk-stabilization exercises. (**A**) Ball sitting with alternate hand and leg raise. (**B**) Bridging on a balance disc, gym ball hold between the knees in supine, prone plank on a balance disc, four-point kneeling with alternate hand and leg raise.

**Figure 3 healthcare-11-02153-f003:**
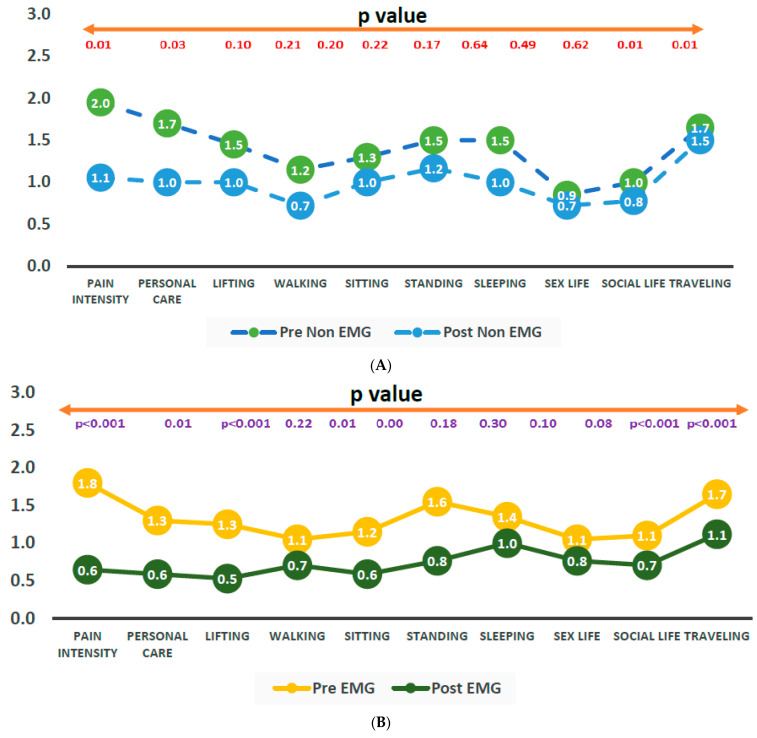
Functional disability as measured using the Oswestry Lower Back Disability Scale in and EMG and non-EMG patients receiving trunk-stabilization exercises. (**A**) Comparison of ODI score in pre-non-EMG and post-non-EMG. (**B**) Comparison of ODI score between pre-EMG and post-EMG. (**C**) Comparison of ODI score between pre-non-EMG and pre-EMG groups.

**Figure 4 healthcare-11-02153-f004:**
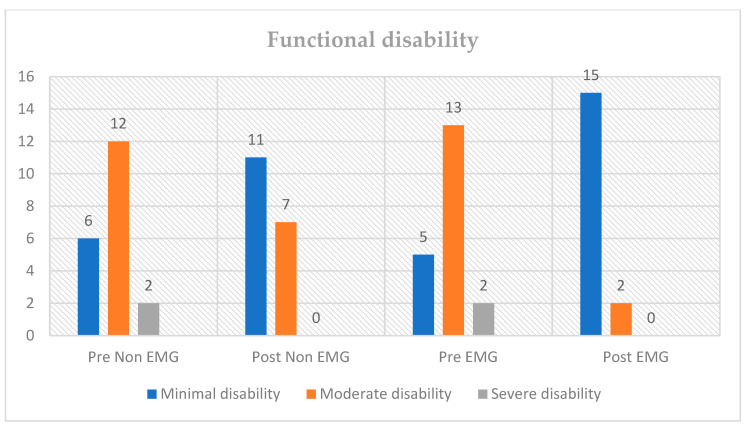
Levels of functional disability based on the ODI score in pre- and post-tests of the EMG and non-EMG groups.

**Table 1 healthcare-11-02153-t001:** Demographic and clinical characteristics of participants in the EMG and non-EMG groups.

Outcome Measures	Non-EMG Group (Mean ± SD)	EMG Group (Mean ± SD)	*t* or χ^2^	*p*
Demographics				
Age (years)	30.45 ± 6.04	30.8 ± 6.7	0.91	0.37
Height (m)	1.59 ± 0.09	1.62 ± 0.08	0.64	0.53
Weight (kg)	60 ± 7.96	63.6 ± 8.88	0.83	0.42
BMI (kg/m^2^)	23.73 ± 3.51	24.17 ± 3.2	0.94	0.36
Marital Status				
Single	1 (5%)	2 (10%)	0.4	0.82
Married	8 (40%)	7 (35%)		
Divorced	1 (5%)	1 (5%)		
Educational Qualification				
Undergraduate	3 (15%)	5 (25%)	0.75	0.69
Postgraduate	5 (25%)	4 (20%)		
Doctorate	1 (5%)	2 (10%)		
Clinical Characteristics				
Duration of NSLBP				
Less than 5 months	3 (15%)	4 (20%)	0.22	0.64
More than 5 months	7 (35%)	6 (30%)		
Medications (NSAIDs)				
No	2 (10%)	3 (15%)	0.61	0.44
Yes	9 (45%)	6 (30%)		
Sleep disturbance				
No	3 (15%)	3 (15%)	0.09	0.77
Yes	6 (30%)	8 (40%)		

χ^2^ = chi square; NSAIDs = nonsteroidal anti-inflammatory drugs.

**Table 2 healthcare-11-02153-t002:** Independent sample test to compare outcome variables such as pain intensity, functional disability, range of motion, and static balance for the same group of cases.

Outcome Measures	Non-EMG Group (Mean ± SD)	EMG Group (Mean ± SD)	*t*	*p*	Cohens d (CI)
Pain					
Pre	5.00 ± 0.79	4.85 ± 0.87	−0.57	0.57	−0.18 (−0.8, 0.44)
Post	2.11 ± 0.67	1.06 ± 0.74	−4.37	*p* < 0.01	−1.48 (−2.22, −0.72)
Functional disability					
Pre (%)	28.1 ± 11.28	26.5 ± 8.07	−0.52	0.61	−0.16 (−0.78, 0.46)
Post (%)	19.89 ± 5.37	14.82 ± 3.08	−3.39	*p* < 0.01	−1.15 (−1.86, −0.42)
Range of Motion					
Flexion					
Pre (cm)	6.38 ± 0.92	5.82 ± 1.77	−1.25	0.22	−0.4 (−1.02, 0.23)
Post (cm)	6.55 ± 0.95	7.61 ± 1.84	2.16	0.04	0.73 (0.04, 1.41)
Extension					
Pre (cm)	3.82 ± 1.09	3.52 ± 1.06	−0.88	0.38	−0.28 (−0.9, 0.35)
Post (cm)	3.91 ± 1.04	4.47 ± 1.21	1.45	0.16	0.49 (−0.19, 1.16)
Lateral Flexion Right					
Pre (cm)	16.4 ± 3.01	15.88 ± 4.06	−0.46	0.65	−0.14 (−0.76, 0.48)
Post (cm)	16.81 ± 2.6	17.14 ± 3.2	0.35	0.73	0.11 (−0.53, 0.76)
Lateral Flexion Left					
Pre (cm)	16.83 ± 3.82	15.78 ± 3.68	−0.89	0.38	−0.28 (−0.9, 0.34)
Post (cm)	17.07 ± 3.7	16.57 ± 3.62	−0.40	0.69	−0.14 (−0.8, 0.53)
Static Balance					
Eyes Open					
Pre	4.84 ± 0.45	4.8 ± 0.56	−0.23	0.82	−0.07 (−0.69, 0.55)
Post	4.87 ± 0.42	5.03 ± 0.41	1.11	0.28	0.37 (−0.3, 1.04)
Eyes Closed					
Pre	4.95 ± 0.5	5.1 ± 0.51	0.94	0.36	0.3 (−0.33, 0.92)
Post	5.06 ± 0.63	5.17 ± 0.44	0.58	0.56	0.2 (−0.47, 0.86)

SD: standard deviation; d: effect size (Cohen’s d); cm: centimeters.

**Table 3 healthcare-11-02153-t003:** Comparison of the pre- and post-test effects of trunk-stability exercises with and without EMG biofeedback in chronic lower back pain patients.

Group	Pre Mean ± SD	Post Mean ± SD	MD	SD	*t*	*p*	95%CI	
							Lower	Upper
Pain								
Non-EMG Group	5.00 ± 0.79	2.11 ± 0.67	2.94	0.63	19.54	*p* < 0.001	2.62	3.26
EMG Group	4.85 ± 0.87	1.06 ± 0.74	3.64	0.86	17.44	*p* < 0.001	3.2	4.09
Functional disability (%)								
Non-EMG Group	28.1 ± 11.28	19.89 ± 5.37	8.33	10.13	3.48	*p* < 0.001	3.29	13.37
EMG Group	26.5 ± 8.07	14.82 ± 3.08	12.58	8.59	6.03	*p* < 0.001	8.16	17
Range of Motion (cm)								
Flexion (cm)								
Non-EMG Group	6.38 ± 0.92	6.55 ± 0.95	−0.1	0.24	−1.73	0.1	−0.22	0.02
EMG Group	5.82 ± 1.77	7.61 ± 1.84	−1.94	2.43	−3.28	*p* < 0.001	−3.19	−0.68
Extension (cm)								
Non-EMG Group	3.82 ± 1.09	3.91 ± 1.04	−0.19	0.54	−1.51	0.14	−0.46	0.07
EMG Group	3.52 ± 1.06	4.47 ± 1.21	−0.88	0.49	−7.3	*p* < 0.001	−1.13	−0.62
Lateral Flexion Right (cm)								
Non-EMG Group	16.4 ± 3.01	16.81 ± 2.6	−0.41	0.93	−1.96	0.06	−0.84	0.02
EMG Group	15.88 ± 4.06	17.14 ± 3.2	−0.57	1.42	−1.64	0.11	−1.3	0.16
Lateral Flexion Left (cm)								
Non-EMG Group	16.83 ± 3.82	17.07 ± 3.7	−0.06	1.03	−0.25	0.8	−0.57	0.45
EMG Group	15.78 ± 3.68	16.57 ± 3.62	−0.3	0.92	−1.33	0.2	−0.77	0.17
Static Balance (cm)								
Eyes Open								
Non-EMG Group	4.84 ± 0.45	4.87 ± 0.42	−0.14	0.33	−1.78	0.09	−0.3	0.02
EMG Group	4.8 ± 0.56	5.03 ± 0.41	−0.1	0.21	−1.99	0.06	−0.2	0.00
Eyes Closed								
Non-EMG Group	4.95 ± 0.5	5.06 ± 0.63	−0.14	0.31	−1.91	0.07	−0.29	0.01
EMG Group	5.1 ± 0.51	5.17 ± 0.44	−0.11	0.18	−2.61	0.01	−0.21	−0.02

MD: mean difference; SD: standard deviation; cm: centimeters.

## Data Availability

The data presented in this study are available on request from the corresponding author. The data are not publicly available due to privacy restrictions.

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
