# Peer review of "Enhancing Functional Ability in Chronic Nonspecific Lower Back Pain: The Impact of EMG-Guided Trunk Stabilization Exercises"

_healthcare, 2023, doi:10.3390/healthcare11152153_

Round 1
Reviewer 1 Report (Previous Reviewer 1)
The adequately authors responded to my previous remarks.
The major previous flaw was study sample size and decreasing number of participants along the study follow-up.
It seems that authors increased the number of participants enrolled into the study. How come in the period of time between submission of 1st version of manuscript and this resubmitted version?
Moreover, I can't find see results at follow-up period, only pre and post.
Figure 2 showing the results at 2 and 4 weeks - what does it mean? Pre, post or follow-up?
Author Response
Respected Sir/mam,
Please see the attached file for the reply to comments suggested from you.
Should you have any further query, I am ready to answer
With regards,

Reviewer 2 Report (Previous Reviewer 2)
Please provide approval number of your ethic committee: that´s ok
Did you register your trial in clinicaltrial.gov? if so, provide number of registration: ok, this is enough explanation
In table 1 provide the effect size (Cohen´s D with 95%CI): -please, merge d and lower and upper 95%CI like this example: 4 (2, 5) and change title column to: Cohen´s D (95%CI). Cohen´s D for age, height, weight and BMI is not neccesary, only for outcome variables, so turn it in blanck. -delete column entitle t
Please provide an initial table 1 with baseline clinical and demographic characteristics of participants with p values to asses the absence of baseline between both groups: that`s ok
Author Response
Respected Sir/mam,
Please see the attached file for the reply to comments suggested from you.
Should you have any further query, I am ready to answer
With regards,

Round 2
Reviewer 1 Report (Previous Reviewer 1)
Authors made appropriate comments and corrections. I found these editing errors:
Figure 1 is seen twice in the text
271 - varidables
272 - FurtherFurther
275 - Table 12 - numbering of following tables is wrong - please check
Author Response
Authors made appropriate comments and corrections.
Thank you for your valuable comments and corrections. We appreciate your input and are glad to see that the changes made align with the intended purpose of the content. Your expertise and attention to detail have helped ensure the accuracy and quality of the material.
I found these editing errors:
Figure 1 is seen twice in the text
Thank you for pointing out the duplication of Figure 1 in the text. We apologize for any confusion it may have caused. Actually this happened as the track change in the file was on. As soon as we stopped the track change in the file, the figure is reflected just once.
271 - varidables
Your feedback is instrumental in improving the overall quality of our work, and we are grateful for your contribution. This is corrected and reflected in the file
272 – FurtherFurther
Again this happened as the track change in the file was on. As soon as we stopped the track change in the file, the term further is written only once and can be seen in the manuscript. Thanks for your valuable suggestion
275 - Table 12 - numbering of following tables is wrong - please check
We reviewed the numbering sequence of the tables and made the necessary corrections to ensure that they are accurately labeled and listed in the correct order. Our aim is to provide readers with clear and organized information, and your feedback is invaluable in achieving that goal.
Sincerely
Corresponding Author

This manuscript is a resubmission of an earlier submission. The following is a list of the peer review reports and author responses from that submission.
Round 1
Reviewer 1 Report
In this research authors investigate the impact of EMG feedback in trunk stabilization exercises in nonspecific low back pain. The study is interesting as there is a big need for evidence in physiotherapy.
My main concerns are on group investigated and study design:
1. Authors in lines 120 – 126 inform that prior power analysis that sample size of 40 participants is necessary. In general or in one group? Moreover, 10 participants were excluded from the study (lines 134-136). Furthermore, only 20 participants were enrolled into study (236-240), which remains unexplained by authors.
2. Study is supposed to be double-blinded. The description of this process remains unclear for me – in lines 196 – 197 “Participants in the EMG group performed trunk stabilization exercise after the electrodes were placed[19, 20]” does in mean that the second group had electrodes but not feedback, or no electrodes. How about the observer?
3. Please provide remarks or studies on validation of the App-Coo-Test used for balance assessment
More remarks:
- - sentence “Nonspecific low back pain is expressed as pain, which is not instigated by a, familiar disease, such as osteoporosis, fracture, structural deformity, infection, tumor, inflammatory condition, radicular or cauda equina syndrome” has some errors or lack some words after “by a,” (lines 47-49)
- - please avoid words or comparisons “like a tiny peable” (line 90)
- - all images are blurred and low quality
- - authors claim and are in favor that EMG is better even if it has advantages only on some parts of evaluation (lines 305-306), range of motion (lines 254-256 – flexion was improved in EMG and extension in non-EMG, but not in EMG group – p=0,08)
- - same goes to conclusion (lines 377-380) – authors write that EMG training helps, but from the study also non-EMG helps significantly eg. in pain, range of motion etc.
- in summary authors should really comment and explain on number of the participants in the study groups, “blindness” of this study and be more balanced when drawing conclusions.
Author Response
Dear sir,
Thank you for your valuable comments and suggestions on our manuscript. We appreciate the time and effort you have dedicated to reviewing our work. Please find attached our detailed response addressing each of your points in a point-wise manner.
If you have any further questions or require additional clarification, please do not hesitate to reach out to us. We are grateful for your guidance and expertise, which will undoubtedly enhance the quality of our manuscript.
Thank you once again for your invaluable input.
with regards

Reviewer 2 Report
Very interesting research, I suggest some issues to improve it:
Please provide approval number of your ethic commintee; did you register your trial en clinicaltrial.gorv? if so, provide number of registration.
In table 1 provide the effect size (Cohen´s D with 95%CI)
Please provide an initia table 1 with baseline clinical and demographic characteristics of participanst with p values to asses de absence of baseline between both groups.
Author Response
Dear Sir,
Thank you for your valuable comments and suggestions on our manuscript. We appreciate the time and effort you have dedicated to reviewing our work. Please find attached our detailed response addressing each of your points in a point-wise manner.
If you have any further questions or require additional clarification, please do not hesitate to reach out to us. We are grateful for your guidance and expertise, which will undoubtedly enhance the quality of our manuscript.
Thank you once again for your invaluable input.
With regards

Reviewer 3 Report
Please find attached.

The manuscript needs to be edited by an English-speaking individual.
Author Response
Dear Sir/Madam,
Thank you for your valuable comments and suggestions on our manuscript. We appreciate the time and effort you have dedicated to reviewing our work. Please find attached our detailed response addressing each of your points in a point-wise manner.
If you have any further questions or require additional clarification, please do not hesitate to reach out to us. We are grateful for your guidance and expertise, which will undoubtedly enhance the quality of our manuscript.
Thank you once again for your invaluable input.
with regards

Round 2
Reviewer 1 Report
Authors answered and clarified most of my doubts and remarks. Unfortunately, tehre is still al problem with the sample size of the group, which is not possible to improve at this stage of the study.
